# The Characterization of the Tobacco-Derived Wild Tomato Mosaic Virus by Employing Its Infectious DNA Clone

**DOI:** 10.3390/biology11101467

**Published:** 2022-10-06

**Authors:** Jinlong Yin, Xin Hong, Sha Luo, Jingquan Tan, Yuanming Zhang, Yanglin Qiu, Muhammad Faizan Latif, Tao Gao, Haijia Yu, Jingke Bai, Shujun Li, Kai Xu

**Affiliations:** 1Jiangsu Key Laboratory for Microbes and Functional Genomics, College of Life Sciences, Nanjing Normal University, Nanjing 210023, China; 2Jiangsu Key Laboratory for Molecular and Medical Biotechnology, College of Life Sciences, Nanjing Normal University, Nanjing 210023, China; 3Key Laboratory for Green Preservation & Control of Tobacco Diseases and Pests in Huanghuai Growing Area, Tobacco Research Institute, Henan Academy of Agricultural Sciences, Xuchang 461000, China

**Keywords:** wild tomato mosaic virus, tobacco, superinfection exclusion, overwinter host

## Abstract

**Simple Summary:**

Wild tomato mosaic virus (WTMV, genus *Potyvirus*, family *Potyviridae*) is an emerging viral pathogen that endangers *Nicotiana tabacum* production. The field survey conducted in this study shows that WTMV is becoming an epidemic in China. An infectious DNA clone of the tobacco-derived WTMV is constructed. It can infect wild eggplant, black nightshade, and tobacco plants but can not infect various local pepper varieties. WTMV evolves into three groups that coincide with their original hosts, tobacco, pepper, or wild eggplant. Thus, the tobacco-derived WTMV might divergently evolves to adapt to tobacco other than peppers. We show that WTMV is compatible with the coinfection of cucumber mosaic virus (CMV) or tobacco mosaic virus (TMV) in tobacco but not other potyviruses. Specifically, WTMV can interfere with the infection of other potyvirus species in tobacco, a phenomenon known as superinfection exclusion previously observed within the same potyviral species. This study contributes essential knowledge on the evolution, infectivity, and recent epidemics of WTMV, and provides the key tool for further disease-resistance and field management studies.

**Abstract:**

Viral diseases of cultivated crops are often caused by virus spillover from wild plants. Tobacco (*N**. tabacum*) is an important economic crop grown globally. The viral pathogens of tobacco are traditional major subjects in virology studies and key considerations in tobacco breeding practices. A positive-strand RNA virus, wild tomato mosaic virus (WTMV), belonging to the genus *potyvirus* in the family *potyviridae* was recently found to infect tobacco in China. In this study, diseased tobacco leaf samples were collected in the Henan Province of China during 2020–2021. Several samples from different locations were identified as WTMV positive. An infectious DNA clone was constructed based on one of the WTMV isolates. By using this clone, we found that WTMV from tobacco could establish infections on natural reservoir hosts, demonstrating a possible route of WTMV spillover and overwintering in the tobacco field. Furthermore, the WTMV infection was found to be accompanied by other tobacco viruses in the field. The co-inoculation experiments indicate the superinfection exclusion (SIE) between WTMV and other *potyvirus* species that infect tobacco. Overall, our work reveals novel aspects of WTMV evolution and infection in tobacco and provides an important tool for further studies of WTMV.

## 1. Introduction

Agriculture started thousands of years ago and provided for the needs of the ever-growing human population. Many plant species were selected to grow in large tracts as a part of human civilization and inevitably changed the ecological conditions of farming areas as well as the traits of the plants themselves. This human practice led to the rapid transmission and outbreak of diseases in crops, including those caused by viruses. Wild plants play a key role in the emergence of viral diseases since they are the natural reservoirs of plant viruses [1]. Almost all the plants can potentially serve as viral reservoirs, but the viral pathogens of crop diseases are usually transmitted from wild plants, often grassy, cruciferous, leguminous, and nightshade weeds [2,3] that are in close taxonomic and geographic relationship to the crops.

Tobacco (*Nicotiana tabacum*) is an economic crop susceptible to many plant viruses. Large-scale tobacco farming started in colonial Colombia in 1778 [4]. The Colombian village of Ambalema was the major tobacco farming center that supplied most of the tobacco export of Colombia in the early 1800s. During that period of time, the tobacco mosaic virus (TMV) became an epidemic in Ambalema [4] and was then transmitted to European countries [5]. TMV can infect various *Nicotiana* species except for *N. glutinosa,* a native of Peru, *N. repanda* of Mexico, *N. rustica* of Ecuador and Peru, and *N. langsdorfii* of Brazil [6]. Some native South American solanaceous species are also resistant to TMV [6,7]. These TMV-resistant plants suggest that they coevolve for the longest period of time with TMV and indicate that the origin of TMV is within the native habitat of these plants [6,7].

Potyviruses are aphid-transmissible positive-strand RNA viruses [8]. Potyviruses that infect tobacco include tobacco etch virus (TEV), potato virus Y (PVY), tobacco vein banding mosaic virus (TVBMV), and tobacco vein mottling virus (TVMV), among which TEV is the most studied one. TEV was first identified in the solanaceous weed *Datura stramonium*, or jimson weed, a tropical America native plant [9], in the Connecticut Agricultural College, USA, in the Autumn of 1915 [10]. It was also found in leguminous weeds, including *Senna tora* (originally *Cassia tora*) and *Senna obtusifolia* (originally *Cassia obtusifolia*), in the USA in 1954 and in Venezuela in 1974 [11,12], and in the nightshade weed *Solanum viarum* in the USA in 1994 [13]. TEV can establish infections in edible eggplant (*Solanum melongena*), pepper (*Capsicum annuum*), and tomato (*Solanum lycopersicum*) [10,14,15,16]. In the tobacco-growing regions of North Carolina, USA, TEV was found to spread during the 1960s and boomed in the early 1980s [16]. TEV is also one of the major prevalent virus types in China [17]. TEV and other potyviruses infecting tobacco can be transmitted by aphids in a non-persistent manner. Thus, weeds or solanaceous vegetables grown near the tobacco planting area can serve as natural hosts or overwintering reservoirs for these viruses.

*Wild tomato mosaic virus* (WTMV) is a *potyvirus* member that was initially identified in wild eggplants (*Solanum torvum*, called wild tomato in the original publication) collected in Laichau of Vietnam from field surveys in the early 2000s [18]. Later, it was found in *S. torvum*, black nightshade (*Solanum nigrum*), *C. annuum* plants, and tobaccos in various locations in China [19,20,21]. WTMV was firstly reported to infect tobacco grown in mid-May 2013 in Nanxiong (Guangdong Province, China), where yellow sun-cured tobacco has been produced for more than 300 years [22]. The existence of WTMV in tobacco fields was later reported in Xichang (Sichuan Province, China) in 2015 [23] and in Xinyang, Nanyang, and Luoyang (Henan Province, China) in 2017 [24]. The WTMV-infected tobacco plants developed yellowing and mosaic symptoms [22,23]. Albeit the full-length genomic sequences of some WTMV isolates derived from different hosts were reported, their phylogenetic relationships remain unclear. Moreover, the infectious DNA clone for WTMV has not been previously reported, and neither was its competitiveness when co-infecting with other tobacco viruses.

In this study, WTMV was detected in the diseased tobacco leaves in Henan Province in 2020 by using the reverse transcription polymerase chain reaction (RT-PCR). The next year, a broader survey revealed several new locations of the WTMV epidemic. To understand the evolutionary history of WTMV, phylogenetic analyses were performed and revealed the divergence of the tobacco-derived WTMV from the other isolates. To explore the virus infectivity, an infectious DNA clone was successfully constructed based on one tobacco-derived WTMV isolate. This clone could establish infection in *Nicotiana tabacum* and *Nicotiana benthamiana* plants and infect the weed hosts, including *S. torvum* and *S. nigrum*. To characterize the coinfection of WTMV and other viruses, co-inoculation experiments were conducted on tobacco. The obtained results demonstrated that tobacco-derived WTMV could compete with the infection of PVY or TEV, and induce more severe symptoms when coinfected with the cucumber mosaic virus (CMV) or promote the co-infecting TMV accumulation. Overall, our work reveals novel aspects of WTMV evolution and infection in tobacco and provides an important tool for further studies of WTMV.

## 2. Materials and Methods

### 2.1. Plants Growth Conditions

The plants were grown in a greenhouse with a temperature of 24 ± 2 °C and photoperiod at 14/10 h (day/night). The light intensity was 7000–10,000 lux, and the relative humidity was maintained at 50–70%. The plant varieties and sources are listed in Appendix A.

### 2.2. Virus Samples COLLECTION and Preservation

The tobacco leaf samples with yellow and mosaic symptoms were collected from fields in Henan Province in 2020 (15 samples) and 2021 (271 samples). They were stored in a refrigerator at −80 °C and transported with dry ice. The detailed locations from where WTMV-positive tobacco samples were collected are listed in Appendix A.

### 2.3. Virus Detection by Western Blot and RT-PCR

The total proteins extracted from the upper leaves of the inoculated plants and samples from the field were separated with 10% sodium dodecyl sulfate-polyacrylamide gel. Then, the proteins were transferred to polyvinylidene fluoride (PVDF) (0.22 μm) membranes for Western blotting. The rabbit polyclonal antibodies against coat proteins of WTMV or PVY, and mouse monoclonal antibodies against coat protein of CMV were used for primary incubation. HRP-conjugated goat anti-rabbit IgG (D110058, Sangon Biotech, Shanghai, China) or goat anti-mouse IgG (D110087, Sangon Biotech, Shanghai, China) was used as secondary antibodies. The peroxidase reaction was conducted with the Chemistar High-sig ECL Western Blotting Substrate kit (180-501, Tanon, Shanghai, China). The PVDF membranes were then stained by ponceau S. The ponceau S or Coomassie brilliant blue stained Rubisco large subunit was used to show protein loading.

For RT-PCR, the total RNAs from the leaves were extracted as previously described [25]. The reverse transcript was conducted with the HiScript III 1st Strand cDNA Synthesis Kit (R312, Vazyme, Nanjing, China). Both Oligo(dT)20VN and Random hexamers were used as primers to initiate cDNA synthesis. The specific primers used for virus detection are listed in Appendix A. The *β-tubulin* gene was amplified as an internal control. The PCR was conducted with Taq DNA Polymerase (P101, Vazyme, Nanjing, China) with the following procedures: 95 °C for 60 s; followed by 35 cycles at 98 °C for 5 s, 58 °C for 15 s, and 72 °C for 60 s. The PCR products were checked in 1% agarose gels.

### 2.4. Infectious Clones Construction

The construction of pCB301-WTMV was performed according to the reported method for infectious clones of potyviruses [25,26,27]. The overlapped virus DNA fragments were amplified using cDNA as the template. A poly(A) tail and an HDVrz sequence were introduced to the end of the last virus fragment by several rounds of over-hang extension PCRs. The overlapping regions between the virus fragments and end of restriction enzymes *Stu* I and *Sma* I linearized vector pCB301-304-CEN [26] were also extended with over-hang extension PCR. Then, the virus fragments and linearized vector were co-transformed into yeast W303-1B for homologous recombination in vivo. The PCR-verified infectious clones extracted from the yeast were used for inoculation experiments. In the case of constructing fluorescent protein-tagged infectious clones, the EGFP or tagRFP coding sequence was amplified from pGD-C-EGFP or pGD-C-tagRFP [28,29] and fused to the P1-HcPro or NIb-CP gene junction site by overlapping extension PCR. In order to release the fluorescent protein from the virus polyprotein, the coding sequence of an additional putative protease recognition site was also introduced to the end of the fluorescent protein coding sequence. The primers used for the infectious clone construction of WTMV, WTMV-P1:GFP-1, WTMV-P1:GFP-2, WTMV-NIb:GFP-1, WTMV-NIb:GFP-2, and WTMV-RFP are listed in Appendix A.

pCB301-TEV-GFP was made using the same method for pCB301-WTMV. Four overlapping fragments were amplified from the TEV-GFP clone [30] using specific primers (Appendix A) and recombined with *Stu* I and *Sma* I linearized vector pCB301-304-CEN [26] in yeast.

### 2.5. Plant Inoculation and Phenotyping

Two methods were used for the inoculation experiments depending on plant species. The agroinfiltration method was used to inoculate *N. benthamiana*, *N. tobaccum*, *C. annuum*, *S. lycopersicum*, and *S. melongena*. *Agrobacterium tumefaciens* EHA105 harboring an infectious clone was suspended in an infiltration solution (10 mM MES [pH 5.6], 10 mM MgCl_2_, and 150 mM acetosyringone), and the concentration was adjusted to OD600 = 1.0. The *A. tumefaciens* EHA105-harboring empty vector pCB301-304-CEN was used as a negative control. The suspensions were infiltrated into the plant leaves with needleless syringes. For the inoculation on the plants of *S. nigrum* and *S. torvum*, the rub inoculation method was adopted. The virus-infected *N. benthamiana* leaves were ground in phosphate-buffered solution (PBS, 0.01 M, pH 7.4) with mortars in a 10:1 (volume of PBS to weight of leaves) ratio to produce the inoculums. Oil paint brushes were used for rub inoculation on the leaves. The plants inoculated with PBS were used as a negative control. About three-week-old seedlings were adopted for inoculation by each method.

Photographs were taken about one-month post-inoculation for tobacco plants and about two-weeks post-inoculation for other species. The infection of fluorescent protein-tagged viruses was tracked with a hand-held lamp LUYOR-3415RG (Luyor, Shanghai, China), and the photographs were taken with an LP510 filter for GFP or a BP590 filter (LUV-590A, Luyor) for RFP. In order to merge the photographs of the plants co-infected by different viruses, Photoshop CC (v14.0) was used to extract the RGB signals of fluorescent proteins.

### 2.6. Phylogenetic Reconstruction

A total of 21 WTMV sequences were collected from the NCBI (National Center for Biotechnology Information) database. The records with 100% identity with any other sequences and the records that shared no common fragment with any other sequences were excluded from further analysis. Sequences of two chilli veinal mottle virus (ChiVMV) isolates and thirteen WTMV isolates were aligned using MAFFT (v7.475) by an auto strategy. The evolutionary history was inferred using the ML and general time reversible models in MEGA11 [31,32]. The tree with the highest log likelihood (−44431.29) was selected. Initial tree(s) for the heuristic search were automatically obtained by applying Neighbor-Join and BioNJ algorithms to a matrix of pairwise distances estimated using the maximum composite likelihood (MCL) approach and then selecting the topology with a superior log likelihood value. A discrete gamma distribution was used to model evolutionary rate differences among the sites (5 categories (+G, parameter = 1.8417)). The rate variation model allowed for some sites to be evolutionarily invariable ([+I], 41.02% sites). The tree was drawn to scale, with branch lengths measured in the number of substitutions per site.

## 3. Results

### 3.1. Genome Sequence and Infectious DNA Clone of Tobacco-Derived WTMV

During the spring and summer of 2020, fifteen diseased tobacco leaf samples with yellow and mosaic symptoms were collected from major tobacco-growing regions in Henan, China. By using RT-PCR with oligo(dT)_18_ and WTMV-specific primers (Appendix A), one sample from Sanmenxia was detected as positive for WTMV infection. In order to construct the infectious DNA clone of this tobacco-derived WTMV isolate, a degenerate primer that hybridized to the 5′-end of the WTMV cDNA and WTMV-specific primers that recognized the most conserved WTMV regions (Appendix A) were used to amplify four overlapping WTMV cDNA fragments. These fragments were sequenced and assembled into a complete genome of the WTMV SMX isolate (Genbank accession #OP169002).

The full-length cDNA of WTMV-SMX was cloned into the shuttle plasmid pCB301-304-CEN via the homologous recombination of the linearized vector and the four viral cDNA fragments in yeast cells [25,26,27]. The generated plasmid, pCB301-WTMV, contains a 35S promoter followed by the DNA sequences coding for WTMV-SMX complete genome, poly(A), and a ribozyme derived from the hepatitis D virus (HDVrz) (Figure 1A). To facilitate tracking the WTMV infection, four versions of the WTMV clones expressing an enhanced green fluorescence protein (EGFP) were constructed. The first two, namely, WTMV-P1:GFP-1 and WTMV-P1:GFP-2, contained an EGFP sequence inserted between the P1 and Hc-Pro coding region and flanked by the P1-HcPro and NIb-CP or 6K1-CI protease cleavage sites (Figure 1B). The third and fourth ones, namely, WTMV-NIb:GFP-1 and WTMV-NIb:GFP-2, had the EGFP flanked by the NIa-NIb protease cleavage site and one of the NIb-CP cleavage sites inserted between the NIb and CP coding regions (Figure 1B). The constructed WTMV clones were inoculated onto *N. benthamiana* leaves via agroinfiltration to assess the infectivity. Two weeks post-inoculation, the upper leaves of inoculated plants were obtained for Western blotting analysis using the polyclonal antibody that specifically recognized the coat protein of WTMV-SMX (Figure 1C). The wild-type WTMV clone successfully rescued the viral infection in *N. benthamiana* plants, as shown by the presence of coat protein in the upper leaves detected by Western blotting. The WTMV-infected plants showed systemic symptoms of deformed leaves and stunted growth (Figure 1D, upper panel). WTMV-NIb:GFP-1 and WTMV-NIb:GFP-2 had similar infectivity as the wild-type WTMV, as shown by the same level of systemic symptoms and CP accumulation, while WTMV-P1:GFP-1 or WTMV-P1:GFP-2 presented weaker or no infectivity (Figure 1C,D). The WTMV-driven expression of EGFP in the infected leaves of WTMV-NIb: GFP-1/2 could be observed under a blue laser-light source, also demonstrating the successful systemic infection of the WTMV clones (Figure 1D).

### 3.2. Evolutionary Relationship of WTMV Isolates

Twelve full-length or partial WTMV genomic sequences were obtained from the Genbank and aligned with that of WTMV-SMX. The position of 8653–9310 nt at the WTMV-SMX genome was successfully aligned with the other WTMV sequences (Appendix A). The aligned sequences covered ~93% (758 out of 813 nt) of the WTMV-SMX coat protein coding region. The evolutionary relationships among the aligned WTMV isolates were then inferred by the construction of a phylogenetic tree using the maximum-likelihood (ML) method (Figure 2). It shows that different WTMV isolates are grouped into different monophyletic clades that mostly match the natural hosts rather than the geographic locations. Clade I includes the WTMV isolated from *S. torvum* in China and Vietnam [18,22] and occupies the basal position, representing an ancestral group. Clade II is represented by two WTMV isolates derived from *C. annuum* in Thailand and China [19]. WTMV-SMX is grouped into clade III, which is mainly branched by WTMV isolates derived from cultivated tobacco. The only exception is the WTMV infecting *S. nigrum* that is grouped with tobacco-infecting WTMV isolates in clade III. Since this *S. nigrum*-infecting WTMV-Sn has the ability to infect tobacco [20], it is likely that *S. nigrum* serves as an intermediate or overwintering host for WTMV to establish an infection in the tobacco field.

### 3.3. Infectivity of WTMV-SMX on Potential Hosts

The natural host of WTMV-SMX is *N. tabacum* tobacco, which is an annual cash crop. The continuous WTMV infection of tobacco requires the ability of WTMV-SMX to infect other perennial plants or weeds growing near the farming field. Thus, whether WTMV-SMX can infect tobacco and *S. nigrum* was tested. The results indicate that WTMV-SMX can establish successful infection on all three tobacco cultivars tested, ‘Samsun NN’, ‘Zhongyan 100’, and ‘Yunyan 87’. The latter two are commercial cultivars that are grown in many major tobacco farming areas. The WTMV-infected tobacco plants showed stunted growth and yellowing and deformed upper leaves (Figure 3A and Appendix A). Western blotting confirmed WTMV accumulation in the diseased upper leaves (Figure 3D, upper panel, and Appendix A). Furthermore, WTMV-NIb:GFP-1 was inoculated onto the seedlings of *S. nigrum*. Successful viral infection can be observed after two weeks, as indicated by the EGFP fluorescence excited under a blue-laser lamp (Figure 3B). Western blotting analysis also confirmed successful WTMV infection (Figure 3D, lower-left panel). Similarly, it was observed that WTMV-NIb:GFP-1 can infect the proposed primitive host *S. torvum* (Figure 3C,D, lower-right panel).

It was reported that *C. annuum* could be infected by WTMV [19], suggesting that the cultivated solanaceous vegetables can serve as intermediate hosts for tobacco-infecting WTMV. To test this possibility, 10 *C. annuum* varieties, 3 *S. lycopersicum* varieties, and 3 *S. melongena* varieties (Appendix A) were grown and inoculated with WTMV. To our surprise, all *C. annuum* and *S. lycopersicum* varieties and one *S. melongena* cultivar ‘Zichangqie’ were unable to host the WTMV infection (Figure 3E and Appendix A). The exception was two southeast Asia *S. melongena* heirloom cultivars, ‘Thai round green’ and ‘Thai lavender frog egg’, which can be grown annually or perennially (Figure 3F and Appendix A). These results show that WTMV-SMX unlikely uses annual vegetable crops, such as *C. annuum* and *S. lycopersicum*, as intermediate hosts.

### 3.4. Field Survey of Major Tobacco Growing Areas in Henan Province of China in 2021

Given the potential harm of the disease caused by WTMV on tobacco farming, to better understand the WTMV epidemics at present, a field survey covering more areas was conducted in Henan Province of China in 2021. Overall, 271 diseased tobacco leaf samples were collected from the major tobacco growing areas, including Xinyang, Zhumadian, Xuchang, Pingdingshan, Luohe, Nanyang, Sanmenxia, Luoyang, Jiyuan, and the experimental station of the Henan Academy of Agricultural Sciences in Xuchang. Twenty-three of them tested positive for WTMV when identified by Western blotting analysis using WTMV CP-specific antibodies (Appendix A). We also tested some WTMV-positive samples in the reaction with TEV and PVY antibodies. We observed that the reactivity of TEV polyclonal antibodies was similar to that of WTMV, while PVY polyclonal antibodies were largely different (Appendix A). These results indicate that the WTMV-positive samples from Western blotting analysis might be infected or accompanied by the infection of other potyviruses. To this end, we subsequently used RT-PCR to verify the presence of WTMV and other common tobacco viruses in these samples. The results show that six samples are verified as positive for WTMV infection (Appendix A), and some of them are co-infected with CMV alone (Figure 4, #41), PVY + TMV (Figure 4, #110), PVY + CMV + TMV (Figure 4, #124), or CMV + TMV (Figure 4, #127). Thus, the natural occurrence of WTMV is often associated with the infection of CMV, TMV, or other potyviruses. The locations of these six samples are Sanmenxia, Xuchang, and Luoyang, of which Xuchang is a new location for the WTMV epidemic in 2021 [24].

### 3.5. Coinfection of WTMV with Other Viruses Reveals SIE between Potyvirus Species

To understand the coinfection of WTMV with other tobacco viruses, the tobacco cv. ‘Zhongyan 100’ plants were agroinfiltrated with WTMV alone or coinfiltrated with CMV [33], PVY-GFP [26], or pCB301-TEV-GFP. Severe mottling and shrinking symptoms of systemically infected leaves were observed in the case of WTMV and CMV coinfection (Figure 5A), suggesting the coinfection of either virus could synergistically enhance the severity of the symptoms. Western blotting of both CMV and WTMV coat proteins of coinfected leaf samples showed that the accumulation of each virus was comparable to that of infection alone (Figure 5C, left panel). In contrast, when WTMV was coinfected with PVY-GFP or TEV-GFP, which also belongs to the *potyvirus* genus, a significant reduction in PVY-GFP or TEV-GFP accumulation can be observed under a blue-laser lamp along with varying levels of symptom changes (Figure 5B). The reduction in PVY-GFP accumulation when co-inoculated with WTMV was also demonstrated by Western blotting analysis. The results show that only WTMV could be detected in the systemically infected leaves (Figure 5C, right panel), suggesting that WTMV is more competitive than PVY-GFP when coinfecting tobacco cv. ‘Zhongyan 100’.

*N. benthamiana* was also used to investigate whether WTMV could promote or inhibit TMV infection since our lab clone of TMV-GFP (30B) [34] is unable to infect tobacco cv. ‘Zhongyan 100’ or ‘Samsun NN’. To visualize WTMV when coinfecting with TMV-GFP during infection, we replaced the GFP coding sequence in WTMV-NIb:GFP-1 with that of tagRFP and generated WTMV-RFP. Subsequently, WTMV-RFP and TMV-GFP were co-inoculated or separately inoculated into *N. benthamiana* plants. This indicates that WTMV-RFP could replicate in cells that were also infected with TMV-GFP (Figure 6A,B). Additionally, TMV-GFP accumulation was enhanced when coinfected with WTMV (Figure 6D, left upper panel). Different from the observation for TMV-GFP coinfection, when PVY-GFP or TEV-GFP were coinfected with WTMV-RFP in *N. benthamiana*, WTMV-RFP-infected upper leaf cells can exclude or be excluded by, PVY-GFP or TEV-GFP as the GFP or RFP fluorescence formed separated cell patches (Figure 6A,B). Immunoblotting analysis confirmed both potyviruses were replicated in the upper leaves (Figure 6D, left lower panel). These results show that the superinfection exclusion (SIE) phenomenon occurs when two members of the *potyvirus* genus coinfect tobacco plants. Similar to TMV-GFP and unlike PVY-GFP or TEV-GFP, the coinfection of CMV did not produce fluorescent patches of WTMV-GFP in systemically infected leaves (Figure 6C). Furthermore, both CMV and WTMV-GFP replicated to similar levels as when they were replicating alone (Figure 6D).

Overall, WTMV infection in tobacco could be compatible with and synergistically enhance the infection of TMV or CMV. However, WTMV competes with the other potyviral infection, as shown by inhibited PVY or TEV accumulation in *N. tabacum* or by the superinfection exclusion phenomenon in *N. benthamiana*.

## 4. Discussion

WTMV is an emerging tobacco virus that was first identified in tobacco in south China in 2013 [22]. Its natural reservoir was proposed to be *S. torvum* or *S. nigrum* [18,20], both of which are solanaceous weeds. *S. torvum* was the first reported natural host of WTMV. Our phylogenetic analysis of the WTMV also suggested that the WTMV isolates infecting tobacco and *C. annuum* evolved from the ancestor, which may also infect *S. torvum*, inferred by the property of the base clade (Figure 2). However, we showed a different evolutionary history for the *S. nigrum*-derived WTMV isolate Sn. WTMV-Sn has a close phylogenetic relationship with WTMV tobacco isolates (Figure 2), suggesting that WTMV-Sn was spilled from tobacco to *S. nigrum* recently or vice versa. *S. nigrum* is a native species in China and can grow as a short-lived perennial. It produces berry-like fruits that are toxic if eaten in large quantities and are traditionally used as a prescription in Chinese medicine. It is possible that *S. nigrum* serves as an intermediate or overwintering host for WTMV to infect crops, such as tobacco, while *S. torvum* serves as a natural reservoir. In agreement with this view, the tobacco-derived isolate WTMV-SMX can replicate in *S. nigrum* and *S. torvum* (Figure 3B,C).

To our surprise, except for two heirloom *S. melongena* varieties native to southeast Asia, the tobacco isolate WTMV-SMX could not infect the overall 14 local solanaceous vegetable varieties, including *C. annuum*, *S. lycopersicum*, and *S. melongena* (Figure 3E,F). This result is different from the successful infection of WTMV in *C. annuum* (19). We assumed that the *C. annuum* varieties used in this study were different from the reported, yet unidentified, variety [19]. Alternatively, WTMV might evolve divergently to adapt to two different hosts, *C. annuum* and tobacco, supported by the phylogenetic analysis (Figure 2). These results guide WTMV management of tobacco, suggesting that more attention should be given to eradicating solanaceous weeds, such as *S. nigrum*, rather than paying attention to nearby vegetable farming.

The epidemic of WTMV in tobacco-growing areas in the Henan Province of China was reported for Xinyang, Nanyang, and Luoyang during the growing season of 2017 [24]. Our survey, mainly based on samples collected in 2021, confirmed that WTMV is still an epidemic in Luoyang. In addition, several new locations were recorded, including Sanmenxia and Xuchang (Appendix A). It is clear that WTMV is gradually spreading in Henan Province.

One intriguing finding in this study was that WTMV infection in tobacco from fields was occasionally accompanied by the coinfection of PVY (Figure 4, #110, and #124). Since potyviruses can be transmitted by aphids, the case of coinfection suggests that the intermediate hosts of PVY and WTMV might be both located close to the tobacco growing field or the intermediate hosts of both viruses are actually the same one. Furthermore, in each case of the coinfection (Figure 4, #110 and #124), only one of the two viruses accumulated to a higher level, suggesting one virus might suppress, or cross protects, the infection of another virus. Our co-inoculation assay of these two viruses in *N. tabacum* and *N. benthamiana* plants well demonstrated the occurrence of SIE between WTMV and PVY, or WTMV and TEV.

It was shown for plum pox virus (PPV), a potyvirus from the plums, that one strain of PPV can interfere with the infection of another strain of the same virus [35]. Similar results were also shown for different strains of potato virus A (PVA) or zucchini yellow mosaic virus [36,37], both of which belong to the genus *potyvirus*. However, different viruses from distinct genera of the *Potyviridae* family can not induce SIE [38]. Different from the above-mentioned studies, we clearly showed that SIE could occur between two species of the *potyvirus* genus. The obtained results also suggest that the mechanism of SIE between different potyviruses is not likely due to the virus-induced RNA silencing as proposed for PVA (37) since WTMV shares low sequence similarity with PVY or TEV.

## 5. Conclusions

In conclusion, we developed a WTMV infectious DNA clone, an important tool for studying WTMV infection. We identified the potential natural reservoir and overwintering host for WTMV and reported that WTMV could interfere with the infection of other potyviruses in tobacco. This work provided a foundation for understanding WTMV epidemics in the tobacco field and will be helpful for future studies of WTMV.

## Figures and Tables

**Figure 1 biology-11-01467-f001:**
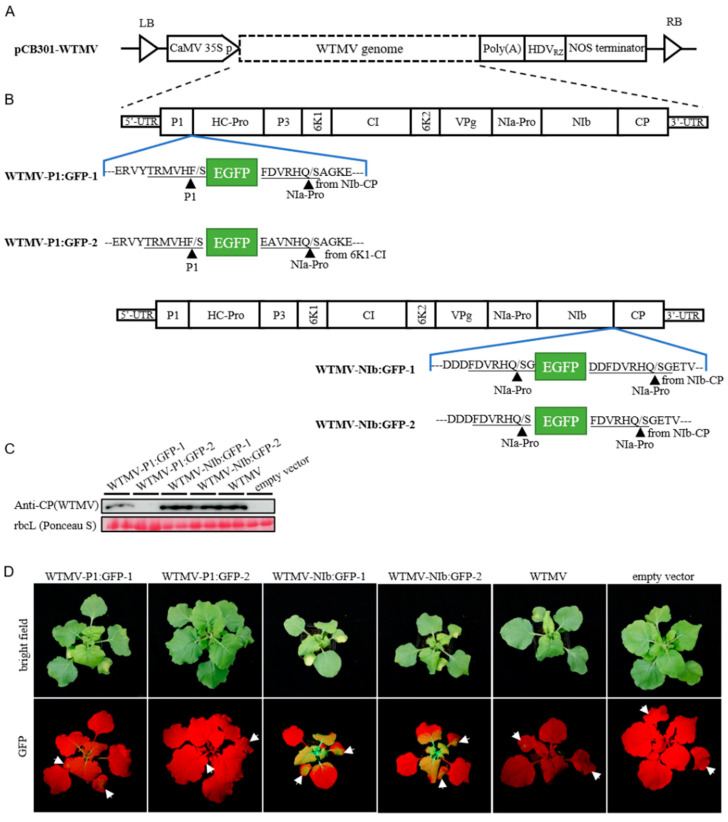
The construction of WTMV infectious clones. (**A**) Schematic representation of the basic WTMV infectious clone. The genomic sequence and hepatitis delta virus ribozyme (HDVrz) are inserted downstream of the CaMV 35S promoter between the left border (LB) and right border (RB) of the binary vector pCB301. (**B**) Schematic representations of different WTMV infectious clones which express an enhanced green fluorescent protein (EGFP). The inverted blue funnel shapes represent the EGFP insertion. The underlined amino acid sequences represent the putative proteases recognize sequences (PRSs). The left PRSs are original, and the right PRSs are duplicated from other regions as indicated. The triangles point to the putative cleavage sites. (**C**) Western blot analysis of the *N. benthamiana* plants agroinfiltrated with WTMV infectious clones. The samples were taken from the upper leaves 2 weeks post-infiltration. Two lanes for each infectious clone indicate biological repeats. Ponceau S stained rubisco large subunit (rbcL) was used to show total protein loading. (**D**) The phenotypes of *N. benthamiana* plants agroinfiltrated with WTMV infectious clones. The photographs are taken 2 weeks post-infiltration. The *N. benthamiana* agroinfiltrated with an empty vector is used as a negative control. Infiltrated leaves are pointed out by arrows. Strong EGFP fluorescence can be observed in the infiltrated and upper leaves of WTMV-NIb:GFP-1/-2-inoculated plants.

**Figure 2 biology-11-01467-f002:**
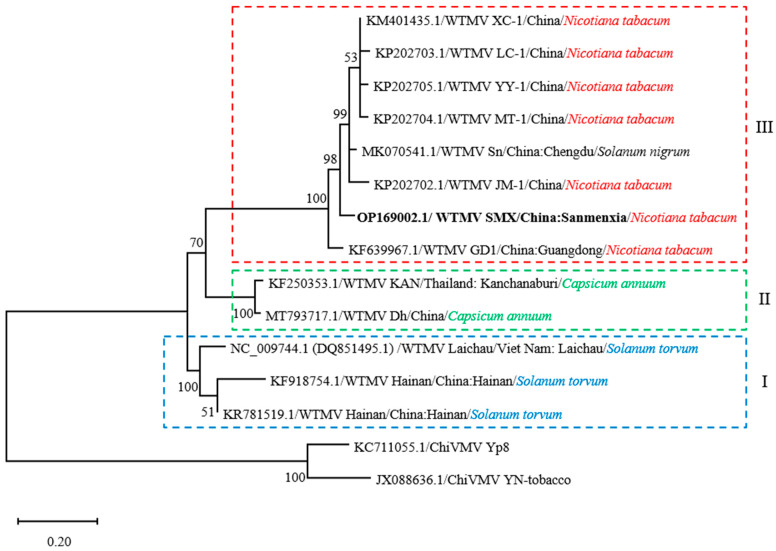
Phylogenetic analyses of the WTMV isolates. The evolutionary history is inferred by using the maximum-likelihood (ML) method. The full-length sequences from NCBI are used for calculations. Two chilli veinal mottle virus (ChiVMV) isolates closely related to WTMV are used as outgroups. Associated branch numbers indicate bootstrap percentages based on 100 replications. The tree is drawn to scale, with branch lengths measured in the number of substitutions per site. Isolates are labeled with Genbank accession numbers, locations, and hosts. The groups based on the phylogeny are emphasized with colored, dashed rectangles.

**Figure 3 biology-11-01467-f003:**
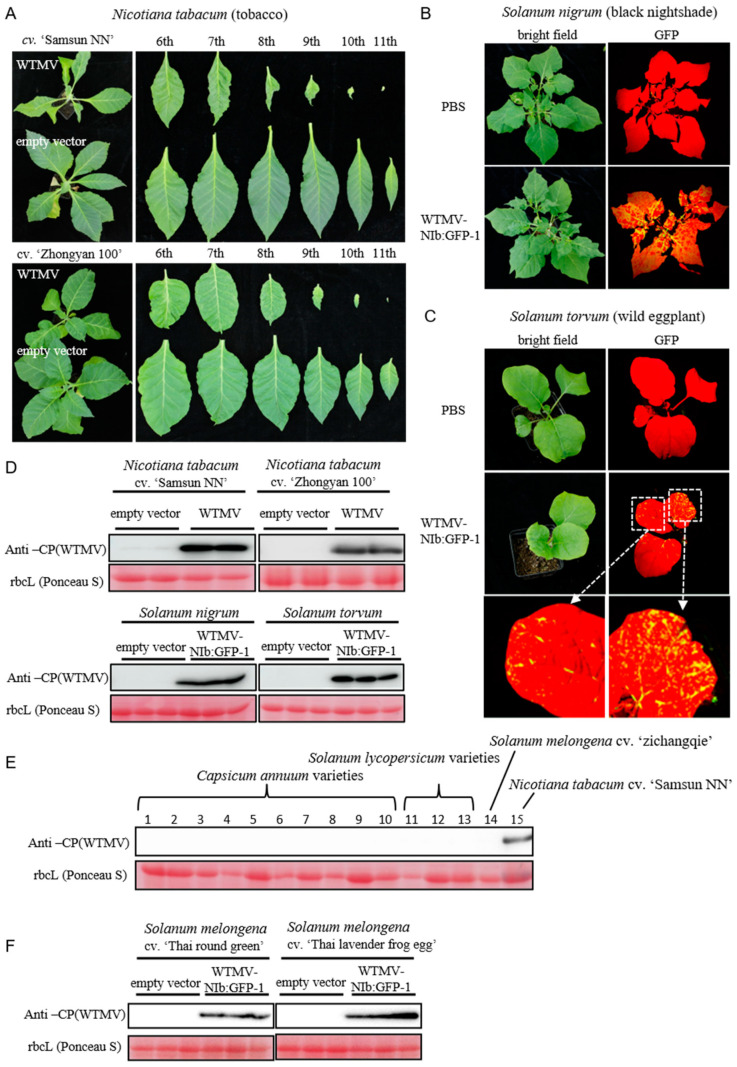
Infectivity of WTMV infectious clone on solanaceous plants. (**A**) The phenotypes of whole plants (left panels) or leaves (right panels) of the *N. tabacum* agroinfiltrated with WTMV infectious clone. The leaf positions are shown. The *N. tabacum* agroinfiltrated with empty vector is used as a negative control. The photographs were taken one-month post-infiltration. (**B**) The phenotypes of *Solanum nigrum* plant rub-inoculated with WTMV-NIb:GFP-1 contain inoculum. The plants inoculated with PBS are used as the control. The photographs were taken 2 weeks post-inoculation. The yellow regions indicate the infection of WTMV. (**C**) The phenotypes of *Solanum torvum* plant rub-inoculated with WTMV-NIb:GFP-1 contain inoculum. See more detail in panel (**B**). (**D**) Western blotting analysis of the plants is described in panels (**A**–**C**). The samples are obtained from the upper leaves of the plants after taking photographs. The different lanes for each treatment indicate biological repeats. (**E**) Western blotting analysis of *Capsicum annuum*, *Solanum lycopersicum*, and *Solanum melongena*, which agroinfiltrated with the WTMV infectious clone. The different lanes for each species indicate different varieties. The sample obtained from WTMV infected *N. tabacum* cv. ‘Samsun NN’ is used as a positive control. (**F**) Western blotting analysis of *S. melongena* cv. ‘Thai round green’ and cv. ‘Thai lavender frog egg’ infected with WTMV-NIb:GFP-1. The samples are obtained from the upper leaves of the plants 2 weeks post-inoculation. The different lanes for each treatment indicate biological repeats. Ponceau S stained rubisco large subunit (rbcL) is used as the loading control.

**Figure 4 biology-11-01467-f004:**
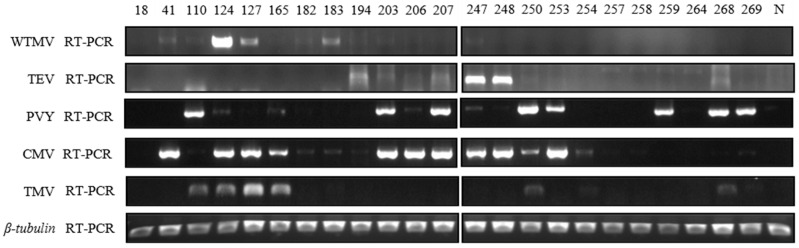
RT-PCR detections of various tobacco viruses in *Nicotiana tabacum* samples collected from fields in Henan Province. The samples are chosen based on the preliminary screen using Western blotting. The sample numbers correspond to the former experiment. The “N” sample was obtained from healthy tobacco and used as a negative control. *β-tubulin* is used as the internal control for RT-PCR.

**Figure 5 biology-11-01467-f005:**
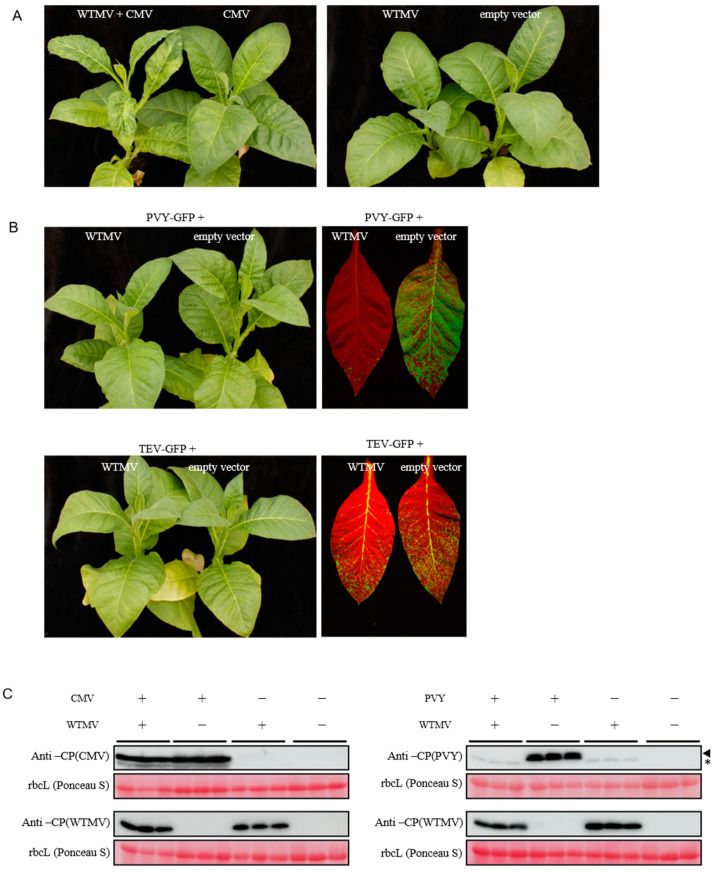
The coinfection of WTMV and other tobacco viruses on *Nicotiana tabacum*. (**A**) The tobacco variety cv. ‘Zhongyan 100’ plants were co-inoculated with WTMV or/and CMV by agroinfiltration. The plants agroinfiltrated with an empty vector were used as a control. (**B**) The tobacco variety cv. ‘Zhongyan 100’ plants were co-inoculated with WTMV or/and another potyvirus as indicated. The photographs of the whole plant and fluorescence images of the infected upper leaf were taken one-month post-infiltration. (**C**) Western blot analysis of the plants described in panels (**A**,**B**). The samples were obtained from the upper leaves after taking the photographs. The different lanes for each treatment indicate biological repeats. Ponceau S stained rubisco large subunit (rbcL) is used as the loading control. The PVY CP-specific band is highlighted by a black triangle. The weaker and lower-molecular-weight band positioned with an asterisk is caused by the unspecific recognition of anti-PVY polyclonal antibodies to the relatively smaller WTMV CP.

**Figure 6 biology-11-01467-f006:**
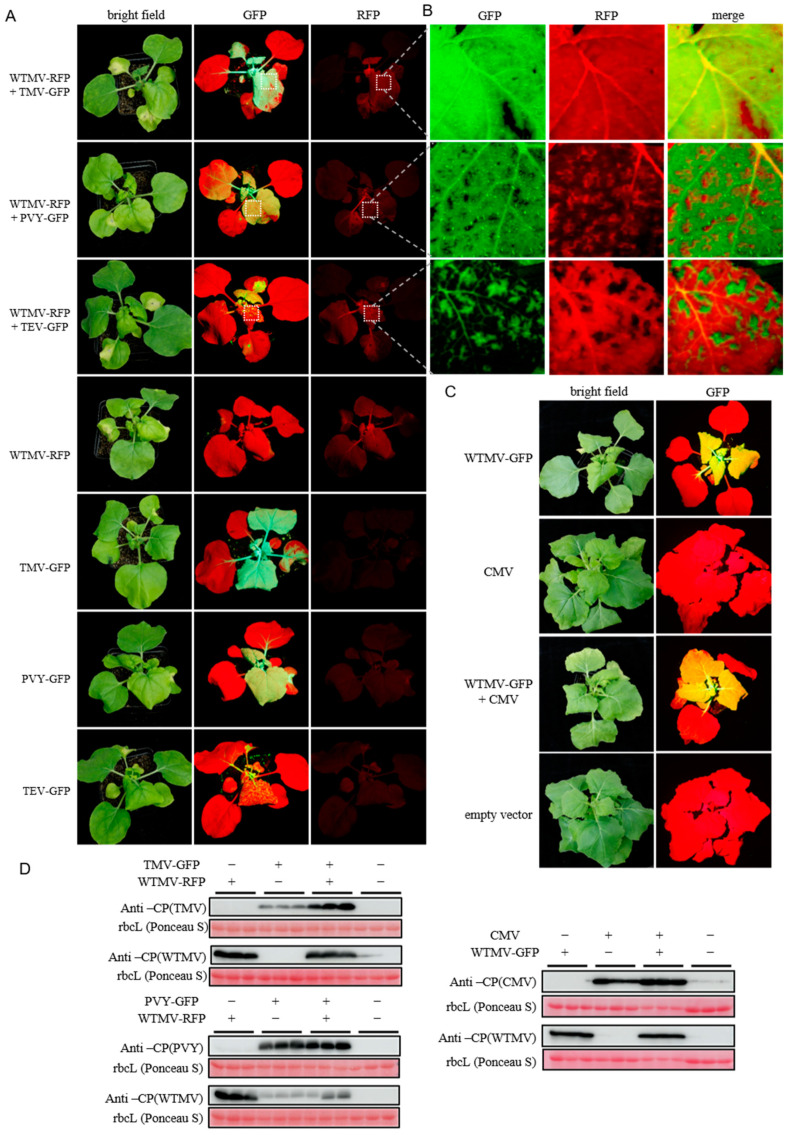
The coinfection of WTMV and other tobacco viruses on *Nicotiana benthamiana*. (**A**,**C**) The *N. benthamiana* plants are co-inoculated by agroinfiltration with different virus infectious clones. The plants infected by single viruses were used as controls. The photographs were taken 2 weeks post-infiltration. (**B**) The zoomed-in views of the leaves from panel (**A**). The green and red channels are extracted from the photographs taken with GFP and RFP filters, respectively. The contrasts are adjusted to minimize the background. (**D**) Western blotting analysis of the plants described in panels (**A**,**C**). The samples were obtained from the upper leaves of the plants after taking photographs. The different lanes for each treatment indicate biological repeats. Ponceau S stained rubisco large subunit (rbcL) is used as the loading control.

## Data Availability

All data are available within the article and its Appendix A.

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
