# Peer review of "The Characterization of the Tobacco-Derived Wild Tomato Mosaic Virus by Employing Its Infectious DNA Clone"

_biology, 2022, doi:10.3390/biology11101467_

Round 1

Reviewer 1 Report

The work is really good with an excellent conceptual background. Some suggessions are given as minor corrections. 

Minor correction:

Line 13: First time give full form of WTMV.

Line 14: Mention the family of WTMV.

Line 20: Full form of CMV or TMV for the first time. 

Line 106: Mention the light intensity and humidity data of the growth environment.

Line 116: 0.22 mm, check the unit.

Line 162: 10× PBS is highly concentrated salt. This concentration is a stress to leaf. Share reason for using high concentrated PBS. 

Line 215: Remove ‘And’ in all starting sentence. 

Fig. 1.C. The number of lanes in rbcL are 12 but the western band number is 6. Mention the reason that two were considered for a specific treatment. 

Fig. 1.D. Give arrow the agroinfiltrated leaf.

Fig. 1.D. Highlight the color chnage between WTMV-NIb:GFP and WTMV-P1:GFP on leaf. 

Fig. 3. E. The rbcL loading is highly variable. 

Fig. 4. The band is not good enough. It has a smearing effect, specially for the tubulin as a control when resolving the amplified product in 1% gel. 

Fig. 5.B. Highlight the difference between WTMT versus Empty vector. 

Fig. 5.C. The rbcL loading is not equal both in Anti-CP (PVY) and Anti-CP (WTMV), The +/+experiment is extremely important to reach a conclusion. 

Reviewer 2 Report

The objective of the paper was found sufficient. However, the writing of the manuscript is complicated and unclear. The results of the study are also unclear. Integrity cannot be achieved and understood at once. There was conceptual confusion. As if the main objectives of the study were achieved with the first-year survey and a sample found infected with WTMV. In 2021, coinfection studies were carried out with 6 infected samples. Therefore, the works do not have continuity with each other like writing. In addition, it seems that the shortcomings of the lack of a virologist subject expert are seen in the study. In my opinion, simplifying the manuscript, and connecting and presenting steps of the study methods and their results can make it more understandable and stronger. In addition to these remarks, I have some comments and suggestions as follows:

1.      The 1st person plural (i.e. we) was used in various places in the manuscript. Instead, it was done; it was observed, it was surveyed; it was screened" should be better to be used.

2.      In the last paragraph of the Introduction, information about the results obtained from the study was given. However, after giving information about the subject of the study and previous studies in the Introduction, it will be sufficient to give what is the aim of the study (Albeit it was mentioned, it was not found adequate), which deficiencies will be resolved, and for what purpose this study was conducted.

3.      In Materials and Methods, the given table 4 was not explanatory and not sufficient for virus samples collection and preservation.

4.      Although it is not essential, terms used such as phenotyping and images in lines 153 and 166 are not very common.

5.      In the manuscript in various lines (55, 63,64, 77, 101, 176, 237, 417, 419, and so on); virus names were written in italic. It should be written in straight not italic and with small letters according to the latest spelling rules of ICTV. That is, if the sentence does not start with the virus name, it should be written as “tomato yellow leaf curl virus”. If the sentence starts with the virus name, it should be written as “Tomato yellow leaf curl virus”.

6.      The explanations in figure 3 are very complex and it takes a lot of work to understand which is A and which is B, C, D, and so on…

7.      Likewise, the statement of supplemental Fig 5 in the description of figure 4 is not very appropriate on page 9.

8.      The explanation of Figure 5 was also not found sufficient and explanatory on page 10.

9.      Why Western Blot was chosen over ELISA? Whereas ELISA is a simpler, cheaper, and faster procedure, less time-consuming, and less hassle for more samples and is more common than Western blot.

10.  In line 307, 23 plant samples were found to be infected with WTMV by western blotting analysis, but in confirmation studies with RT-PCR performed afterward, six samples were found to be positive for WTMV infection. This finding is indeed confusing finding. Although RT-PCR is a much more sensitive test than western blotting, the risk of getting an erroneous result of approximately 75% with western blot causes serious confusion. In reality, the probability of detecting the virus even in samples that do not show symptoms with RT-PCR is much higher than in western blotting, and it could not be understood that samples infected with western blotting were found to be negative by RT-PCR.

11.  It was also stated on line 309 states that 6 samples were found to be positive, but Table 4 shows 7 instead of 6.

12.  Figure 4A was mentioned in the manuscript on lines 315, 316, and 317, but 4A was not in the fig subtitle.

13.  It is thought that the expression WTMV disease on line 394 is used incorrectly. WTMV is a viral organism. It is not the name of the disease it causes.

14.  Why the surveys were carried out more in 2021 and not homogeneously as it was said in lines 395-396.

15.  The expression "different viruses from different Potyvirus genera of the Potyviridae family" in line 420 cannot be understood.

Reviewer 3 Report

Wild tomato mosaic virus is an emerging viral pathogen that endangers Nicotiana tabacum production. In the manuscript,a infectious cDNA clone of wild tomato mosaic virus with GFP gene was developped,and infectivity of WTMV-SMX on potential hosts were analyzed using this infectious clone. In addition, superinfection exclusion phenomenon was also observed when WTMV mix-infected with other potyviral specie. Therefore, I think this paper deserves to be published. Two minor problems are as follows:

1. As the author says, "This result is different from the successful infection of WTMV in C. annuum (19)", one possible reason was the different variety used in the study. The variety information is important for studying the pathogenicity of one virus. Suggesting that the authors add the relative variety information to the used tobacco and others.

2. Line 309-310:  what samples were detected by RT-PCR? How about the  consistency of the detection between WB and RT-PCR?Please rephrase the sentence “ The results showed that six samples were verified as positive for WTMV infection (Supplemental Table 4)”make it more clearly.

Round 2

Reviewer 2 Report

Thank you very much for considering our suggestions and opinions.  I wish you a more successful scientific life.